# Instrument Detection and Descriptive Gesture Segmentation on a Robotic Surgical Maneuvers Dataset

Irene Rivas-Blanco *, Carmen López-Casado, Juan M. Herrera-López, José Cabrera-Villa and Carlos J. Pérez-del-Pulgar

Institute for Mechatronics Engineering and Cyber-Physical Systems (IMECH.UMA), University of Málaga, Andalucía Tech, 29070 Málaga, Spain; mclopezc@uma.es (C.L.-C.); juanma_hl@uma.es (J.M.H.-L.); carlosperez@uma.es (C.J.P.-d.-P.)
* Correspondence: irivas@uma.es

**Abstract:** Large datasets play a crucial role in the progression of surgical robotics, facilitating advancements in the fields of surgical task recognition and automation. Moreover, public datasets enable the comparative analysis of various algorithms and methodologies, thereby assessing their effectiveness and performance. The ROSMA (Robotics Surgical Maneuvers) dataset provides 206 trials of common surgical training tasks performed with the da Vinci Research Kit (dVRK). In this work, we extend the ROSMA dataset with two annotated subsets: ROSMAT24, which contains bounding box annotations for instrument detection, and ROSMAG40, which contains high and low-level gesture annotations. We propose an annotation method that provides independent labels for the right-handed tools and the left-handed tools. For instrument identification, we validate our proposal with a YOLOv4 model in two experimental scenarios. We demonstrate the generalization capabilities of the network to detect instruments in unseen scenarios. On the other hand, for gesture segmentation, we propose two label categories: high-level annotations that describe gestures at a maneuvers level, and low-level annotations that describe gestures at a fine-grain level. To validate this proposal, we have designed a recurrent neural network based on a bidirectional long-short term memory layer. We present results for four cross-validation experimental setups, reaching up to a 77.35% mAP.

**Keywords:** robotic dataset; instrument detection; gesture segmentation; surgical robotics

## 1. Introduction

Surgical data sciences is emerging as a novel domain within healthcare, particularly in the field of surgery. This discipline holds the promise of significant advancements in various areas such as virtual coaching, assessment of surgeon proficiency, and learning complex surgical tasks through robotic systems [1]. Additionally, it contributes to the field of gesture recognition [2,3]. Understanding surgical scenes has become critical to the development of intelligent systems that can effectively collaborate with surgeons during live procedures [4]. The availability of extensive datasets related to the execution of surgical tasks using robotic systems would support these advancements forward, providing detailed information of the surgeon's movements with both kinematics and dynamics data, complemented by video recordings. Furthermore, public datasets facilitate the comparison of different algorithms proposed in the scientific literature. Rivas-Blanco et al. [5] provide a list of 13 publicly accessible datasets within the surgical domain, such as Cholec80 or M2CAI16 datasets.

While most of these datasets include video data [6,7], only two of them incorporate kinematic data [8,9], which is fundamental in analyzing metrics associated with the tool motion. Kinematic data are provided by research platforms such as the da Vinci Research Kit (dVRK). The dVRK is a robotic platform based on the first-generation commercial da Vinci Surgical System (by Intuitive Surgical, Inc., Sunnyvale, CA, USA). This platform offers a software package that records kinematics and dynamics data of both the master tool and

the patient side manipulators. The JIGSAWS dataset [8] and the UCL dVRK dataset [9] records were acquired using this platform. The first one stands out as one of the most renowned dataset in surgical robotics. It encompasses 76-dimensional kinematic data in conjunction with video data for 101 trials of 3 fundamental surgical tasks (suturing, knot-tying, and needle-passing) performed by 6 surgeons. The UCL dVRK dataset comprises 14 videos using the dVRK across 5 distinct types of animal tissue. Each video frame is associated with an image of the virtual tools produced using a dVRK simulator.

Common annotations in surgical robotics datasets are related to tool detection and gesture recognition. These annotations, i.e., labeling data with relevant tags to make it easier for computers to understand and interpret images, are the basis for developing new strategies to advance in the field of intelligent surgical robots and surgical scene understanding. One of the most studied applications in this field is surgical image analysis. There are many promising works for object recognition based on surgical images. Most works perform surgical instrument classification [10,11], instruments segmentation [12,13], and tools detection [7,14]. Al-Hajj et al. [15] propose a network that concatenates several CNN layers to extract visual features of the images and RNNs for analyzing temporal dependencies. With this approach, they are able to classify seven different tools in cholecystectomy surgeries with a performance of around 98%. Sarikaya et al. [7] applied a region proposal network with a multimodal convolutional one for instrument detection, achieving a mean average precision of 90%. Besides surgical instruments, anatomical structures are an essential part of the surgical scene. Thus, organ segmentation provides rich information for understanding surgical procedures. Liver segmentation has been addressed by Nazir et al. [16] and Fu et al. [17] with promising results.

Another important application in surgical data sciences is surgical task analysis, as the basis for developing context-aware systems or autonomous surgical robots. In this domain, the recognition of surgical phases has been extensively studied, as it enables computer-assisted systems to track the progression of a procedure. This task involves breaking down a procedure into distinct phases and training the system to identify which phase corresponds to a given image. Petscharnig and Schöffmann [18] explored phase classification on gynecologic videos annotated with 14 semantic classes. Twinanda et al. [19] introduced a novel CNN architecture, EndoNet, which effectively performs phase recognition and tool presence detection concurrently, relying solely on visual information. They demonstrated the generalization of their work with two different datasets. Other researchers delve deeper into surgical tasks by analyzing surgical gestures instead of phases. Here, tasks such as suturing are decomposed into a series of gestures [8]. This entails a more challenging problem as gestures exhibit greater similarity to each other compared to broader phases. Gesture segmentation, so far, has primarily focused on the suturing task [20–22], with most attempts conducted in in vitro environments yielding promising results. Only one study [23] has explored live suturing gesture segmentation. Trajectory segmentation offers another avenue for a detailed analysis of surgical instrument motion [24,25]. It involves breaking trajectories into sub-trajectories, facilitating learning from demonstrations, skill assessment, phase recognition, among other applications. Authors take advantage of kinematics information provided by surgical robots, which combined with video data, allows more accurate results [26,27].

In our previous work [28], we presented the Robotic Surgical Maneuvers (ROSMA) dataset. This dataset contains kinematic and video data for 206 trials of 3 common training surgical tasks performed with the dVRK. In the work of Rivas-Blanco et al. [28], we presented a detailed description of the data and the recording methodology, along with the protocol of the three performed tasks. This first version of ROSMA did not include annotations. In the present work, we incorporate manual annotations for surgical tool detection and gesture segmentation. The main novelty regarding the annotation methodology is that we have annotated independently the two instruments handled by the surgeon, i.e., we provide metadata with the bounding box position for the tool handled with the right master tool manipulator, and also for the one handled with the left one. We believe that

this distinction would provide useful information for supervisory or autonomous systems, making it possible to pay attention to one particular tool, i.e., one may be interested in the position of the main tool or in the position of the support tool, even if they are both the same type of instrument.

In most of the state-of-the-art works, phases, such as suturing, or gestures, such as pulling suture or needle orientation, are considered an action that involves both tools. However, the effectiveness of these methods relies heavily on achieving a consensus on the surgical procedure, which limits their applicability. Consequently, the generalization to other procedures remains poor. Thus, for the gesture annotations presented in this paper, the approach of treating the two tools independently is also followed. We have annotated each video frame with a gesture label for the right-handed tool and another one for the left-handed tool. This kind of annotating gestures makes special sense for tasks that may be performed with either the right or the left hand, as is the case of the ROSMA dataset, in which tasks are performed for half of the trials with the right hand, and the other half with the left hand. Being able to detect the basic action each tool is performing would allow the identification of general gestures or phases either for dexterous or left-handed surgeons.

In summary, the main contributions of this letter are as follows:

1. This work completes the ROSMA dataset with surgical tool detection and gesture annotations, resulting in 2 new datasets: ROSMAG40, which contains 40 videos labeled with the instruments' gestures, and ROSMAT24, which provides bounding box annotations of the instruments' tip for 24 videos of the original ROSMA dataset.
2. Unlike previous work, annotations are performed on the right tool and on the left tool independently.
3. Annotations for gesture recognition have been evaluated using a recurrent neural network based on a bi-directional long short-term memory layer, using an experimental setup-up with four cross-validation schemes.
4. Annotations for surgical tool detection have been evaluated with a YOLOV4 network using two experimental setups.

## 2. Materials and Methods

### 2.1. System Description

The dVRK, supported by the Intuitive Foundation (Sunnyvale, CA, USA), arose as a community effort to support research in the field of telerobotic surgery [29]. This platform is made up of hardware of the first-generation da Vinci system along with motor controllers and a software framework integrated with the Robot Operating System (ROS) [30]. There are over thirty dVRK platforms distributed in ten countries around the world. In this work, we have used the dVRK platform located at The Biorobotics Institute of the Scuola Superiore Sant'Anna (Pisa, Itally). This platform has two Patient Side Manipulators (PSM), labelled as PSM1 and PSM2 (Figure 1a), and a master console consisting of two Master Tool Manipulators (MTM), labelled as MTML and MTMR (Figure 1b). MTMR controls PSM1, while MTML controls PSM2. For the experiments described in this paper, the stereo vision is provided using two commercial webcams, as the dVRK used for the experiments was not equipped with the endoscopic camera manipulator.

Each PSM has 6 joints following the kinematics described in [31], and an additional degree of freedom for opening and closing the gripper. The tip of the instrument moves around a remote center of motion, where the origin of the base frame of each manipulator is set. The motion of each manipulator is described by its corresponding *base_frame* with respect to the the common frame *ECM*, as shown in Figure 2a. The MTMs used to remotely teleoperate the PSMs have 7-DOF, coupled with the capability to open and close the instrument. The base frames of each manipulator are related through the common frame *HRSV*, as shown in Figure 2b. The transformation between the base frames and the common one in both sides of the dVRK is described in the json configuration file that can be found in the ROSMA Github repository (https://github.com/SurgicalRoboticsUMA/rosma_dataset, accessed on 26 February 2024).

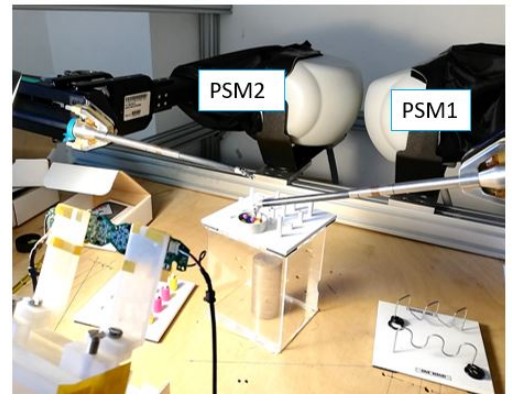

(**a**) Slave side

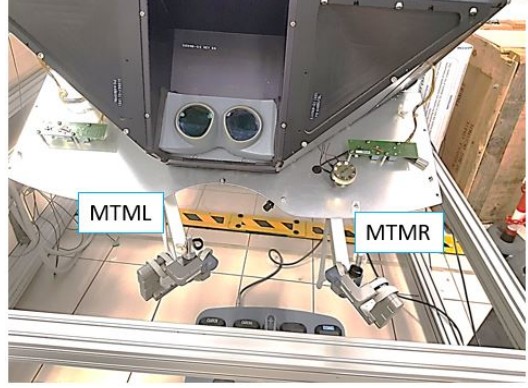

(**b**) Master console

**Figure 1.** da Vinci Research Kit (dVRK) platform used to collect the ROSMA dataset. (**a**) The slave side has two Patient Side Manipulators (PSM1 and PSM2), two commercial webcams to provide stereo vision and to record the images, and the training task board. (**b**) The master console has two Master Tool Manipulators (MTML and MTMR) and a stereo vision system.

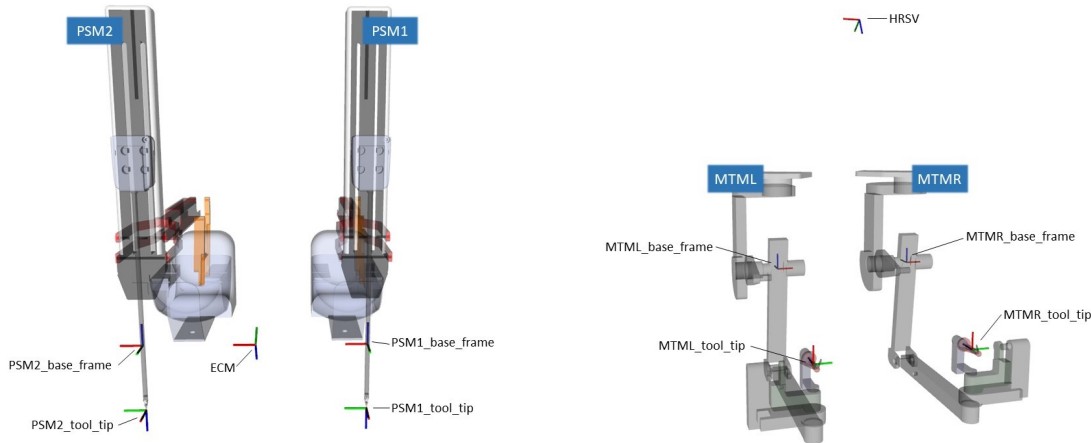

(**a**) Patient side kinematics

(**b**) Surgeon side kinematics

**Figure 2.** Patient and surgeon side kinematics. Kinematics of each PSM is defined with respect to the common frame ECM, while the MTMs are described with respect to frame HRSV.

### 2.2. Robotics Surgical Maneuvers Dataset (ROSMA)

The Robotics Surgical Maneuvers (ROSMA) dataset is a large dataset collected using the platform described in the previous section. This dataset contains the performance of three tasks of the Skill-Building Task Set (from 3-D Technical Services, Franklin, OH, USA): *post and sleeve*, *pea on a peg*, and *wire chaser*. These training platforms for clinical skill development provide challenges that require motions and skills used in laparoscopic surgery, such as hand-eye coordination, bimanual dexterity, depth perception, or interaction between dominant and non-dominant hands [32]. A detailed description of these tasks is described in Section 2.3.

The ROSMA dataset contains 36 kinematic variables, divided into 154-dimensional data, recorded at 50 Hz for 206 trials of three common training surgical tasks. A detailed description of these kinematic data is provided in Table 1. This table provides the kinematic variables recorded along with the number of features recorded for each component. These data are complemented with the video recordings collected at 15 frames per second with $1024 \times 768$ pixel resolution. The dataset also includes task evaluation annotations based on time and task-specific errors, a synchronization data file between data and videos, the transformation matrix between the camera and the PSMs, and a questionnaire with

personal data of the subjects (gender, age, dominant hand) and previous experience using teleoperated systems and visuo-motor skills (sport and musical instruments). This dataset is fully described by Rivas-Blanco et al. [28] and publicly available for download at Zenodo [33]. The code for using the data in Matlab and ROS can be found in Appendix A.

**Table 1.** Description of the kinematic data.

| Kinematic Variable | Component | No. Features |
|---|---|---|
| Cartesian position (x, y, z) | PSM1 | 3 |
| | PSM2 | 3 |
| | MTML | 3 |
| | MTMR | 3 |
| Orientation (x, y, z, w) | PSM1 | 4 |
| | PSM2 | 4 |
| | MTML | 4 |
| | MTMR | 4 |
| Linear velocity (x, y, z) | PSM1 | 3 |
| | PSM2 | 3 |
| | MTML | 3 |
| | MTMR | 3 |
| Angular velocity (x, y, z) | PSM1 | 3 |
| | PSM2 | 3 |
| | MTML | 3 |
| | MTMR | 3 |
| Wrench force (x, y, z) | PSM1 | 3 |
| | PSM2 | 3 |
| | MTML | 3 |
| | MTMR | 3 |
| Wrench torque (x, y, z) | PSM1 | 3 |
| | PSM2 | 3 |
| | MTML | 3 |
| | MTMR | 3 |
| Joint position | PSM1 | 7 |
| | PSM2 | 7 |
| | MTML | 6 |
| | MTMR | 6 |
| Joint velocity | PSM1 | 7 |
| | PSM2 | 7 |
| | MTML | 6 |
| | MTMR | 6 |
| Joint effort | PSM1 | 7 |
| | PSM2 | 7 |
| | MTML | 6 |
| | MTMR | 6 |

## 2.3. Gesture Annotations

In most of the works on gesture segmentation in surgical procedures, gestures are considered an action or maneuver that involves the motion of two surgical instruments, such as suturing, knot-tying, inserting the needle in the tissue, etc. Usually, these kinds of maneuvers are performed following a similar protocol, where actions are always performed by the same tool, i.e., complex tasks that require high precision are usually performed with the right-handed tool, while the left-handed one is usually employed for support tasks. However, left-handed surgeons may not follow this convention, as their dexterous hand is the left. Thus, we believe that being able to recognize gestures that are defined by an adjective instead of by an action would facilitate the generalization of the gesture segmentation algorithms to different protocols of the same task. As well, the generalization

to different ways of performing a particular task, for example, in the case of left-handed surgeons, would also be easier.

As mentioned previously, the ROSMA dataset contains the performance of three tasks: *post and sleeve*, *pea on a peg*, and *wire chaser*. To explore the idea of gesture segmentation presented above, in this work we have only annotated the two first tasks of the dataset, as these two tasks have similar procedure protocols, which allow us to have common gestures for both of them. However, *wire chaser* has a very different protocol, so we cannot define the same gestures as for the other two tasks. *Post and sleeve* task (Figure 3a) consists of moving the colored sleeves from one side of the board to the other. Each user performs six trials: three starting from the right side and the other three starting from the left side. On the other hand, *pea on a peg* (Figure 3b) consists of picking six beads from the cup and placing them on top of the pegs. As in the previous task, each user performs six trials: three placing the beads on the left-side pegs, and the other three on the opposite pegs. The detailed protocol of these tasks is described in Table 2. Although these two tasks have different procedures, they follow the same philosophy: picking an object and placing it on a peg. Hence, the idea behind the gesture annotation methodology of this work is being able to recognize the gesture or actions regardless of whether the user is performing *pea on a peg* or *post and sleeve*.

**Table 2.** Protocol of the task *pea on a peg* and *post and sleeve* of the ROSMA dataset.

| | Post and Sleeve | Pea on a Peg |
|---|---|---|
| Goal | To move the colored sleeves from side-to-side of the board. | To put the beads on the 14 pegs of the board. |
| Starting position | The board is placed with the peg rows in a vertical position (from left to right: 4-2-2-4). The six sleeves are positioned over the six pegs on one of the sides of the board. | All beads are on the cup. |
| Procedure | The subject has to take a sleeve with one hand, pass it to the other hand, and place it over a peg on the opposite side of the board. If a sleeve is dropped, it is considered a penalty and it cannot be taken back. | The subject has to take the beads one by one out of the cup and place them on top of the pegs. For the trials performed with the right hand, the beads are placed on the right side of the board, and vice versa. If a bead is dropped, it is considered a penalty and it cannot be taken back. |
| Repetitions | Six trials: three from right to left, and other three from left to right. | Six trials: three placing the beads on the pegs of the right side of the board, and the other three on the left side. |

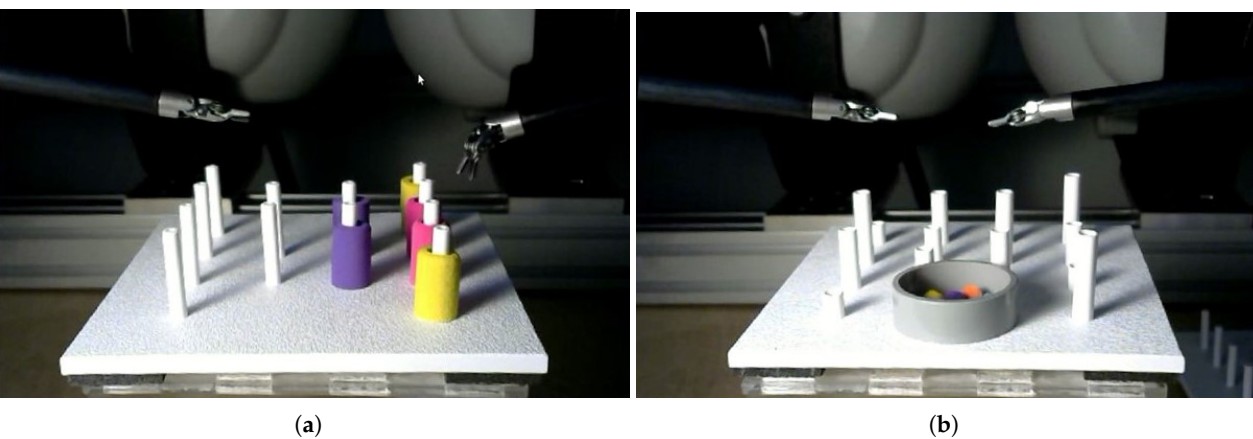

(a)          (b)

**Figure 3.** Experimental board scenario for the ROSMA datasets tasks: (**a**) post and sleeve and (**b**) pea on a peg.

### 2.3.1. ROSMAG40 Annotations

ROSMAG40 is a subset of ROSMA dataset that includes gesture annotations for 40 videos (72,843 frames). The distribution of the annotated videos according to the user, the task, and the dominant hand of each trial is described in Table 3. ROSMAG40 includes videos from five different users: users *X01*, *X02*, and *X07* are dexterous, while users *X06*, and *X08* are ambidextrous. For each user, we have annotated eight video, four for the *pea on a peg* task, and four for *post and sleeve*. For each task, we have annotated two trials performed using PSM1 as the dominant tool and other two trials using PSM2 as the dominant tool (for *pea on a peg*; the dominant tool is the one in charge of picking and placing the peas, while for *post and sleeve*, the dominant tool is considered the one that picks the sleeves).

**Table 3.** Description of the ROSMAG40 dataset distribution.

| User ID | Task | Dominant Tool | Annotated Videos |
|---|---|---|---|
| X01 | Pea on a Peg | PSM1 | 2 |
| | | PSM2 | 2 |
| | Post and sleeve | PSM1 | 2 |
| | | PSM2 | 2 |
| X02 | Pea on a Peg | PSM1 | 2 |
| | | PSM2 | 2 |
| | Post and sleeve | PSM1 | 2 |
| | | PSM2 | 2 |
| X06 | Pea on a Peg | PSM1 | 2 |
| | | PSM2 | 2 |
| | Post and sleeve | PSM1 | 2 |
| | | PSM2 | 2 |
| X07 | Pea on a Peg | PSM1 | 2 |
| | | PSM2 | 2 |
| | Post and sleeve | PSM1 | 2 |
| | | PSM2 | 2 |
| X08 | Pea on a Peg | PSM1 | 2 |
| | | PSM2 | 2 |
| | Post and sleeve | PSM1 | 2 |
| | | PSM2 | 2 |

For the annotations, we have followed the previous idea of defining the gestures to facilitate their generalization to other tasks. In this sense, we define a descriptive gesture as a gesture or action characterized by an action adjective. For example, we can define a descriptive gesture as a *precision action*, regardless of whether we are inserting the needle in a particular point, putting a staple, or dissecting a duct. This work defines two classes of descriptive gestures: maneuver descriptors (MD) and fine-grain descriptors (FGD). Maneuver descriptors represent high-level actions that are common to all surgical tasks, such as precision or collaborative actions. On the other hand, fine-grain descriptors represent low-level actions that are specifically performed in the ROSMA training tasks, such as picking or placing, but which are not generalizable for other types of tasks. ROSMAG40 provides manual annotations for these two classes of gestures. This dataset is available for download at the Zenodo website [34], and its directory structure is as follows:

- FGDlabels: this folder contains text files with the FGD annotations. Each row, which corresponds with a video frame, contains two items: the gesture label for PSM1 and the gesture label for PSM2.
- MDlabels: this folder contains text files with the MD annotations with the same structure as FGDlabels files.
- Kinematics: this folder contains text files with the 34 kinematic features. The first row of these files contains the name of the kinematic feature corresponding with each column.

- Video: this folder contains the video files in mp4 format for the 40 trials of the tasks.

2.3.2. Maneuver Descriptor (MD) Gestures

MD gestures describe the gestures at a higher level than FGD and represent gestures that are common to most surgical tasks. We have defined four MD gestures:

- *Idle (G1)*: the instrument is in a resting position.
- *Precision (G2)*: this gesture is characterized by actions that require an accurate motion of the tool, such as picking or placing objects.
- *Displacement (G3)*: this gesture is characterized by motions that do not require high accuracy, i.e., displacement of the tools either to carry an object or to position the tip in a particular area of the scenario.
- *Collaboration (G4)*: both instruments are collaborating on the same task. For *pea on a peg*, collaboration occurs when the dominant tool needs support, usually to release peas held together by static friction, so it is an intermittent and unpredictable action. For *post and sleeve*, collaboration is a mandatory step between picking and placing an object, in which the sleeve is passed from one tool to another.

Table 4 summarizes the gesture description, along with their ID, labels, and the presence (number of frames) of each gesture in the dataset for PSM1 and PSM2.

**Table 4.** Description of ROSMAG40 annotations for MD gestures.

| Gesture ID | Gesture Label | Gesture Description | No. Frames PSM1 | No. Frames PSM2 |
| --- | --- | --- | --- | --- |
| G1 | Idle | The instrument is in a resting position | 28,395 (38.98%) | 2583 (35.46%) |
| G2 | Precision | The instrument is performing an action that requires an accurate motion of the tip. | 9062 (12.43%) | 9630 (13.1%) |
| G3 | Displacement | The instrument is moving with or without an object on the tip | 24,871 (34.14%) | 26,865 (36.42%) |
| G4 | Collaboration | Both instruments are collaborating on the same task. | 10,515 (14.53%) | 10,515 (14.53%) |

Figure 4 shows six characteristic snapshots of each MD gesture, three for *pea on a top* (top) and three for *post and sleeve* (bottom images). Figure 5 shows a box chart for the MD gestures occurrence for the 40 videos of the dataset. These plots clearly show the dispersion of the data for different trials of the tasks, which was otherwise expected due to the non-uniform nature of the different trials (half of the trials were performed with PSM1 as the dominant tool, and the other half with PSM2). Besides this fact, there is a non-uniform workflow between both tasks:

- *Pea on a peg* is mostly performed with the dominant tool, which follows the flow of picking a pea and placing it on top of a peg. The other tool mainly carries out collaborative actions to provide support for releasing peas from the tool. Thus, the dominant tool gestures follow mostly the following flow: *displacement (G3)-precision (G2)-displacement (G3)-precision (G2)*, with short interruptions for *collaboration (G4)*. While the other tool is mainly in an *idle (G1)* position, with some interruptions for *collaboration (G4)*. This workflow is shown in Figure 6, which represents the sequential distribution of the gestures along a complete trial of the task.
- *Post and sleeve* tasks follow a more sequential workflow between the tools: one tool picks a sleeve and transfers it to the other tool, which places it over a peg on the

opposite side of the board. Thus, the workflow is more similar for both tools, as can be seen in Figure 7.

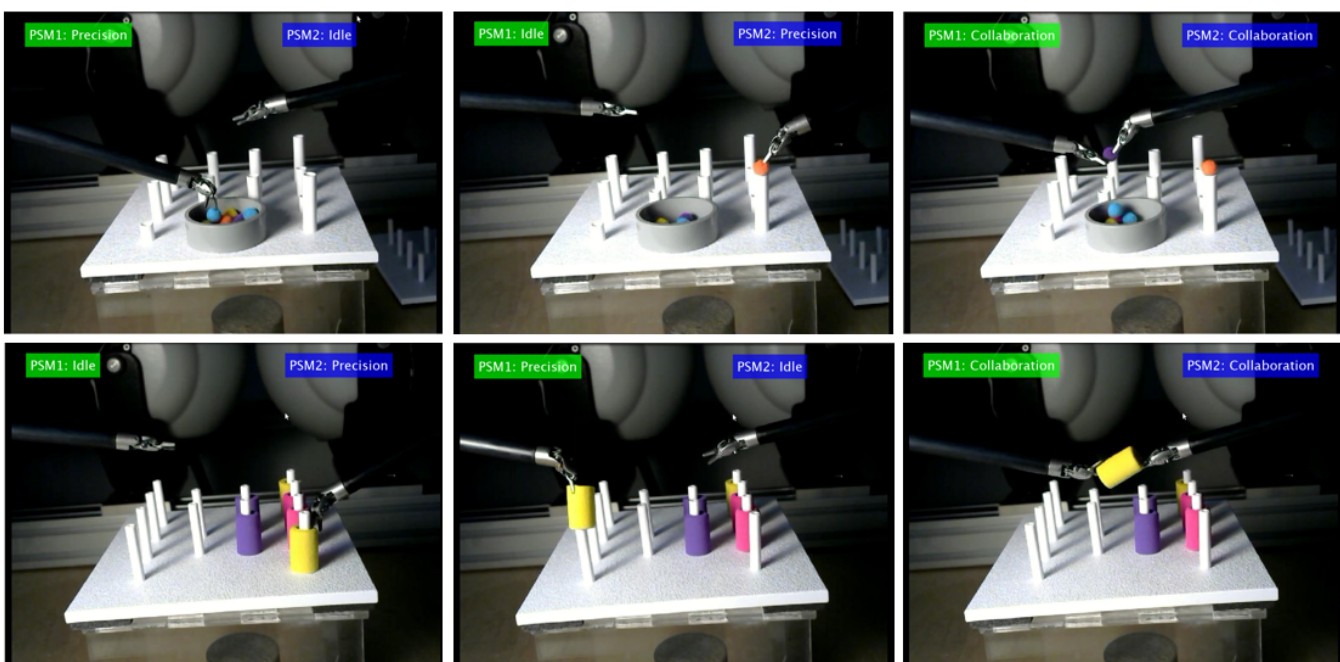

**Figure 4.** Examples of MD gesture annotations.

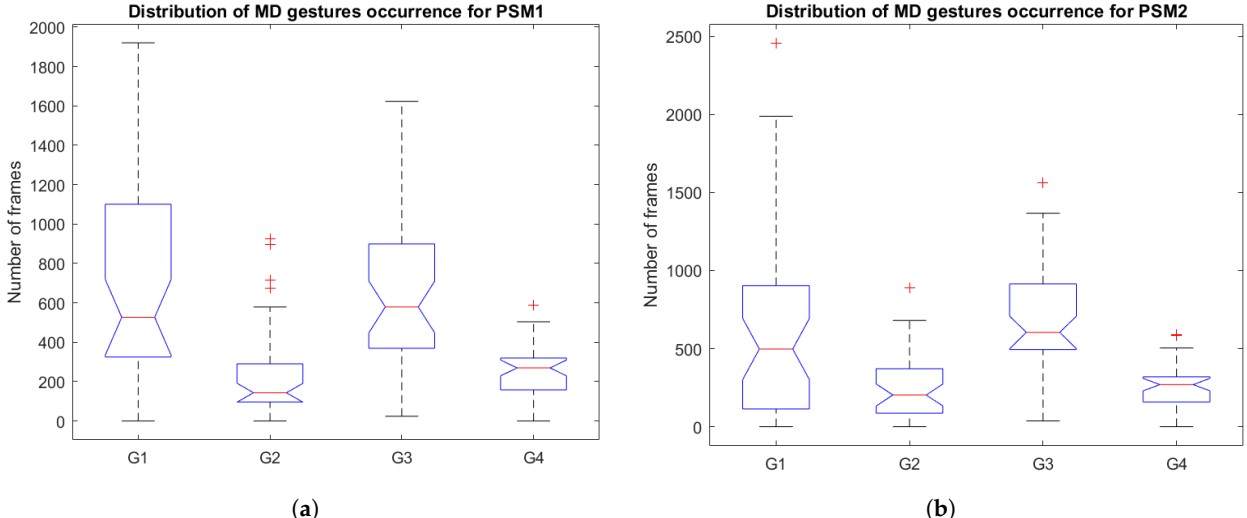

**Figure 5.** Distribution of the gestures occurrence for the MD gestures for (**a**) PSM1 and (**b**) PSM2. The red central mark on each box indicates the median, and bottom and top edges indicate the 25th and the 75th percentiles, respectively. The whiskers extend to the most extreme data points not considered outliers, and the outliers are plotted individually using the '+' marker symbol.

### 2.3.3. Fine-Grain Descriptor (FGD) Gestures

FGD gestures describe the gestures at lower level, i.e., the adjectives used to define the gesture are linked to specific actions of the tools. We have defined six FGD gestures, which are common for the two tasks of the dataset. Most of them can be seen as a particularity of a MD gesture; the relationship between them is shown in Table 5. Additionally, Table 6 presents the ID, label and description of each gesture, along with the number of frames annotated for PSM1 and PSM2. These six FGD gestures are as follows:

- *Idle (F1)*: this is the same gesture as for MD gestures described in the previous section (*G1*).
- *Picking (F2)*: the instrument is picking an object, either a pea on the *pea on a peg* task or a colored sleeve on the *post and sleeve* task. This gesture is a particularity of *G2* of MD descriptors.
- *Placing (F3)*: the instrument is placing an object, either a pea on top of a peg or a sleeve over a peg. This gesture is also a particularity of *G2*.
- *Free motion (F4)*: the instrument is moving without carrying anything at the tip. This gesture corresponds with actions of approaching the objective to pick, and it is a particularity of *G3* of MD descriptors.
- *Load motion (F5)*: the instrument moves while holding an object. This gesture corresponds with actions of approaching the objective to a place, and, therefore, it is also a particularity of *G3*.
- *Collaboration (F6)*: equivalent to gesture *G4* of maneuver descriptors.

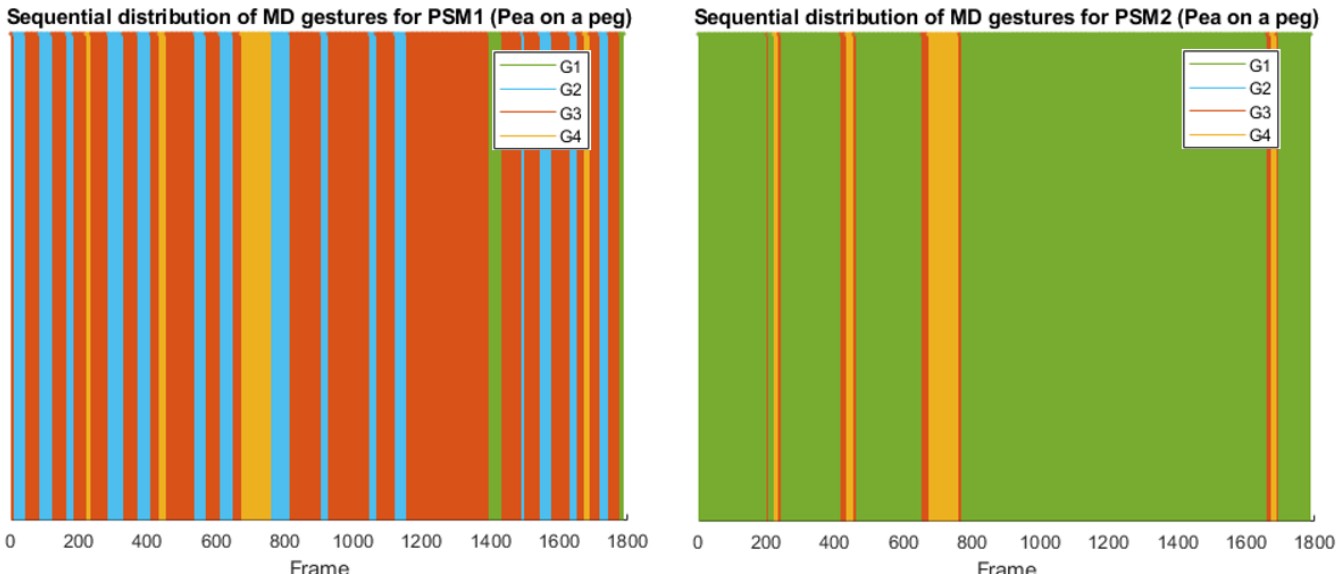

**Figure 6.** Sequential distribution of the MD gestures along a complete trial of a pea on a peg task for PSM1 (**left**) and PSM2 (**right**).

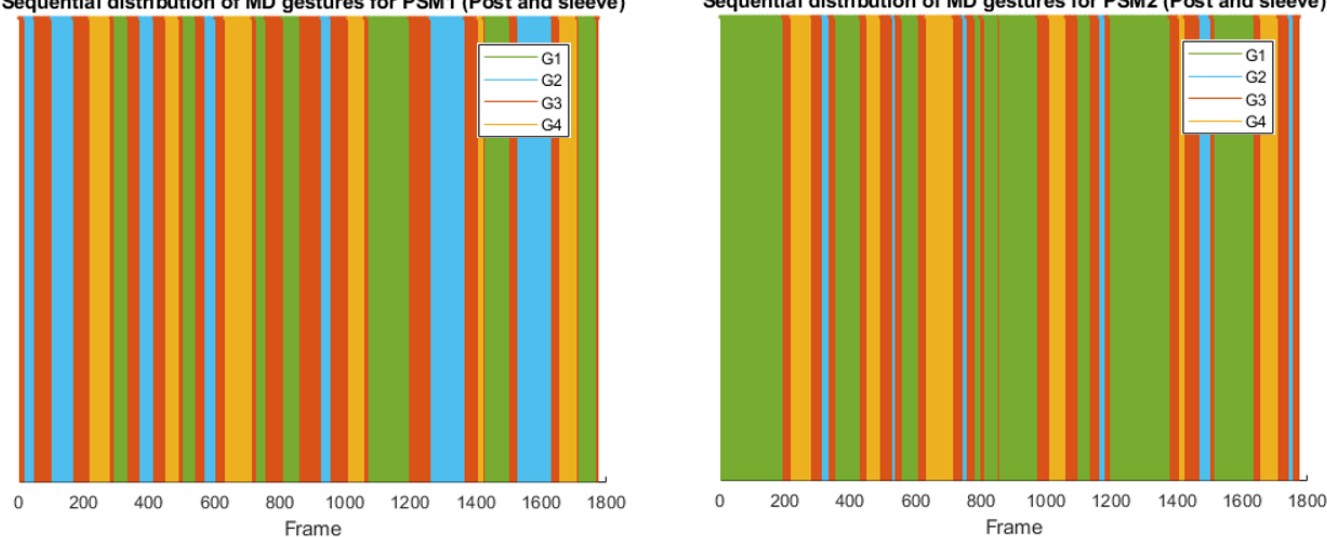

**Figure 7.** Sequential distribution of the MD gestures along a complete trial of post and sleeve task for PSM1 (**left**) and PSM2 (**right**).

**Table 5.** Relation between MD and FGD annotations.

| MD Gestures | FGD Gestures | Description |
|---|---|---|
| Idle (G1) | Idle (F1) | Resting position |
| Precision (G2) | Picking (F2) Placing (F3) | Picking an object Placing an object |
| Displacement (G3) | Free Motion (F4) Load Motion (F5) | Moving without an object Moving with an object |
| Collaboration (G4) | Collaboration (F6) | Instrument collaborating |

**Table 6.** ROSMAG40 annotations for maneuver descriptor gestures.

| Gesture ID | Gesture Label | Gesture Description | Number of Frames PSM1 | Number of Frames PSM2 |
|---|---|---|---|---|
| F1 | Idle | The instrument is in a resting position | 28,395 (38.98%) | 25,830 (35.46%) |
| F2 | Picking | The instrument is picking an object. | 3499 (4.8%) | 4287 (5.8%) |
| F3 | Placing | The instrument is placing an object on a peg. | 5563 (7.63%) | 5343 (7.3%) |
| F4 | Free motion | The instrument is moving without carrying anything at the tool tip. | 15,813 (21.71%) | 16,019 (21.99%) |
| F5 | Load motion | The instrument moves while holding an object. | 9058 (12.43%) | 10,846 (14.43%) |
| F6 | Collaboration | Both instruments are collaborating on the same task. | 10,515 (14.53%) | 10,515 (14.53%) |

Figure 8 shows six characteristic snapshots of each FGD gesture, three for *pea on a peg* (top) and three for *post and sleeve* (bottom images). Figure 9 shows the gesture occurrence distribution for FGD gestures. As expected, this box chart reveals the same conclusions of MD gestures in term of the dispersion of the data for the different task trials. This can also be seen in Figures 10 and 11, which show the sequential workflow of the gestures for a pea on a peg task and a post and sleeve task, respectively:

- The workflow of *pea on a peg* task for FGD gesture is mainly as follows: *free motion (F4)-picking (F2)-load motion (F5)-placing (F3)*, with short interruptions for *collaboration (F6)*. The other tool is mostly in an *idle (F1)* position, with some interruptions for *collaboration (F6)*.
- As we stated previously, *post and sleeve* tasks follow a more sequential workflow between the tools. In the trial represented in Figure 11, PSM1 was the dominant tool, so the comparison between the gesture sequential distribution for PSM1 and PSM2 reflects that *picking* is a more time-consuming task than *placing*.

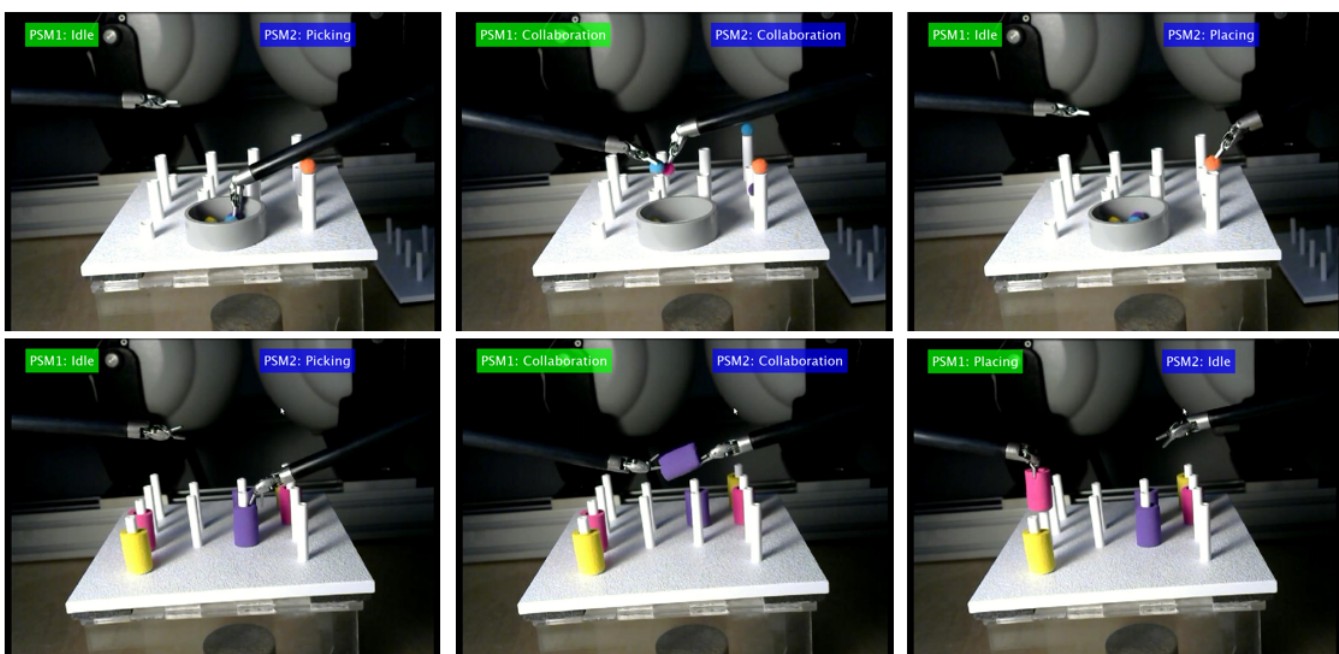

**Figure 8.** Examples of FGD gesture annotations.

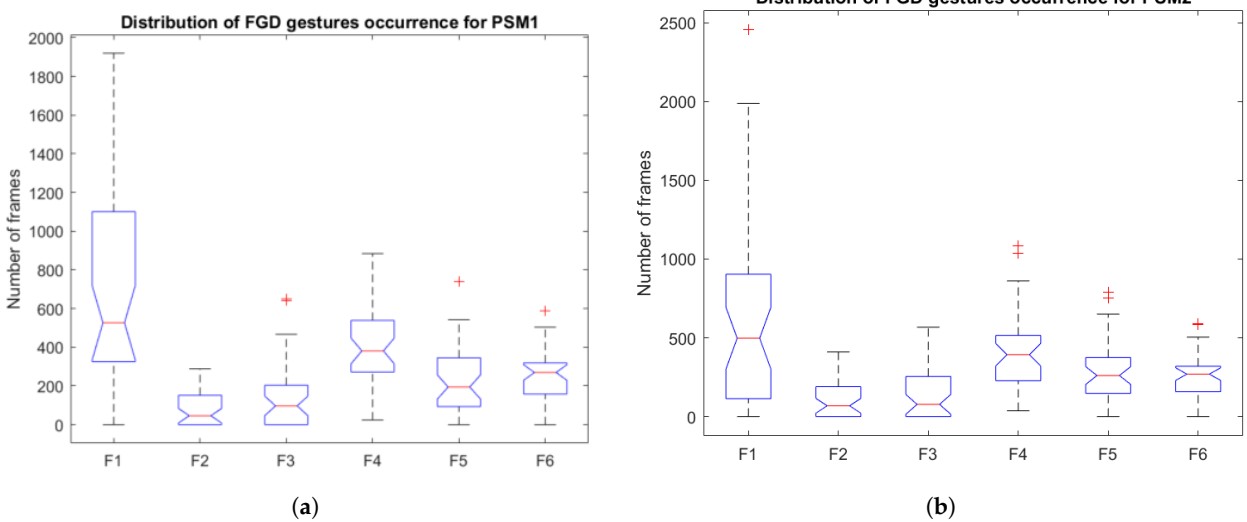

| (**a**) | (**b**) |
|---|---|

**Figure 9.** Distribution of the occurrence for the FGD gestures for (**a**) PSM1 and (**b**) PSM2. The red central mark on each box indicates the median, and bottom and top edges indicate the 25th and the 75th percentiles, respectively. The whiskers extend to the most extreme data points not considered outliers, and the outliers are plotted individually using the '+' marker symbol.

### 2.4. Instruments Annotations (ROSMAT24)

In this work, we extend the usability of ROSMA dataset incorporating manual annotations for instrument detection. Hence, we present the ROSMAT24 dataset, a subset of ROSMA that includes bounding box annotations for instruments detection on 24 videos, 22 videos of *pea on a peg* instances, and 2 videos of *post and sleeve*. Unlike most of the previous work on instrument detection, we provide separate labeled bounding boxes for the tip of PSM1 and PMS2. This way, we can model a network able to distinguish between both tools. We have annotated a total of 48,919 images: 45,018 images (92%) of *pea on a peg* trials, and 3901 (8%) of *post and sleeve*. The idea of this non-uniform distribution of the annotations between the tasks is to validate the robustness of the recognition method for different scenarios. Table 7 shows the specific trials that have been annotated and the

overall number of labeled frames. The videos have been manually annotated frame by frame using the Matlab Toolbox *Video Labeler*. The bounding boxes surround the robotic tip of the tools, which is easily identifiable by its grey color. As the tools have an articulated tip, the size of the bounding boxed depends on the orientation of the tool. Figure 12 shows two examples of the instruments bounding boxes annotations. This dataset is available for download at the Zenodo website [35]. The dataset directory structure is as follows:

- Labels: this folder contains the text files with the bounding boxes for the tip of PSM1 and PSM2. Each row, which corresponds with a video frame, has the following eight items:

$$[BX_1, BY_1, BW_1, BH_1, BX_2, BY_2, BW_2, BH_2] \tag{1}$$

where *BX*, *BY*, *BW*, and *BH* are the coordinates of the bounding boxes x1 (left), y1 (top), width and height, respectively, and the Subindexes 1 and 2 refer to PSM1 and PSM2, respectively.
- Video: this folder contains the video files in mp4 format.

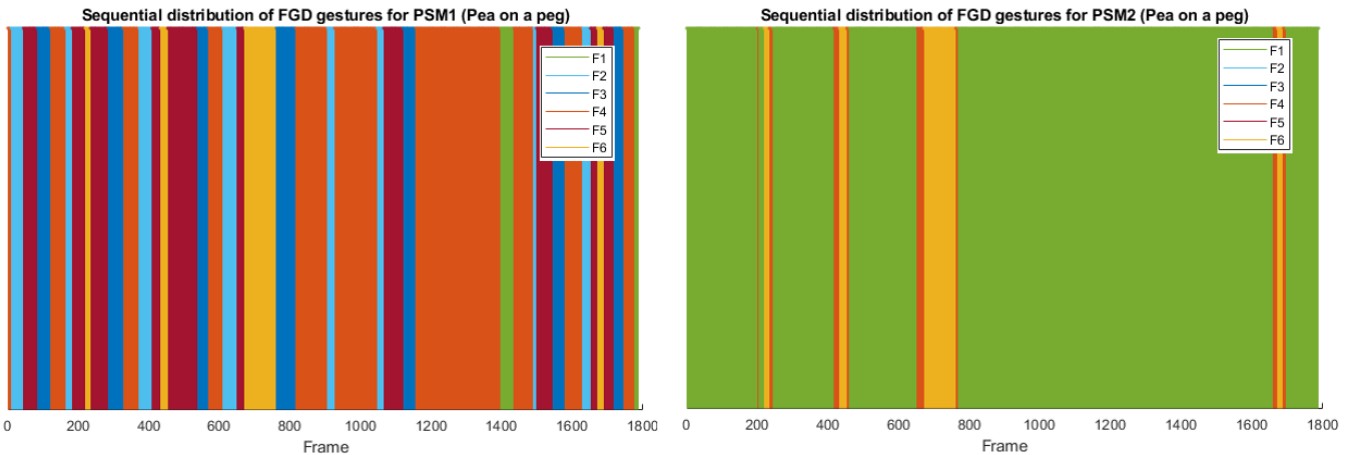

**Figure 10.** Sequential distribution of the FGD gestures along a complete trial of a pea on a peg task for PSM1 (**left**) and PSM2 (**right**).

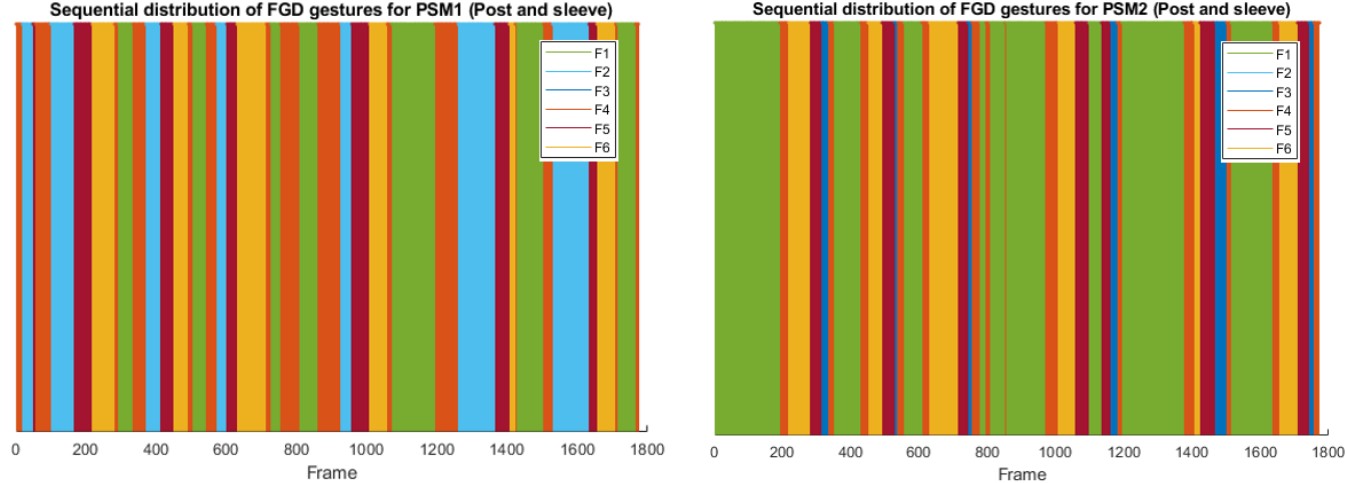

**Figure 11.** Sequential distribution of the FGD gestures along a complete trial of post and sleeve task for PSM1 (**left**) and PSM2 (**right**).

**Table 7.** Description of the ROSMAT24 dataset annotations.

| Video | No. Frames | Video | No. Frames |
|---|---|---|---|
| X01 Pea on a Peg 01 | 1856 | X03 Pea on a Peg 01 | 1909 |
| X01 Pea on a Peg 02 | 1532 | X03 Pea on a Peg 02 | 1691 |
| X01 Pea on a Peg 03 | 1748 | X03 Pea on a Peg 03 | 1899 |
| X01 Pea on a Peg 04 | 1407 | X03 Pea on a Peg 04 | 2631 |
| X01 Pea on a Peg 05 | 1778 | X03 Pea on a Peg 05 | 1587 |
| X01 Pea on a Peg 06 | 2040 | X03 Pea on a Peg 06 | 2303 |
| X02 Pea on a Peg 01 | 2250 | X04 Pea on a Peg 01 | 2892 |
| X02 Pea on a Peg 02 | 2151 | X04 Pea on a Peg 02 | 1858 |
| X02 Pea on a Peg 03 | 1733 | X04 Pea on a Peg 03 | 2905 |
| X02 Pea on a Peg 04 | 2640 | X04 Pea on a Peg 04 | 2265 |
| X02 Pea on a Peg 05 | 1615 | X01 Post and Sleeve 01 | 1911 |
| X02 Pea on a Peg 06 | 2328 | X11 Post and Sleeve 04 | 1990 |

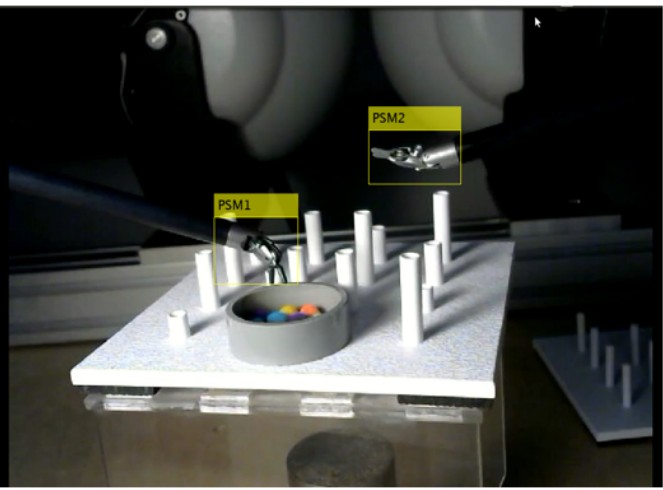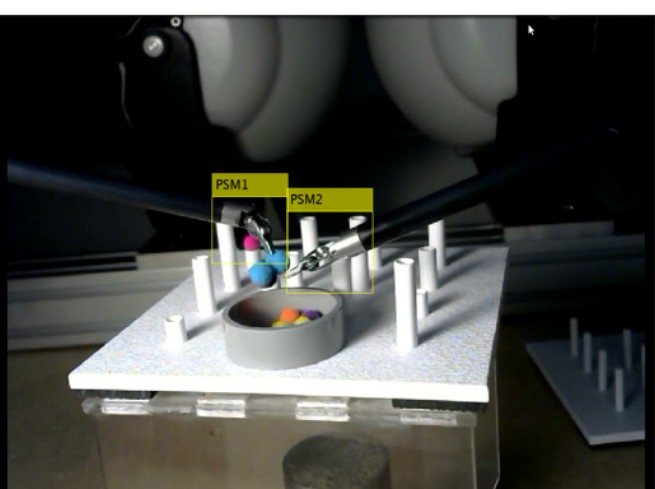

**Figure 12.** Examples of the instruments bounding boxes annotations of ROSMAT24.

*2.5. Evaluation Method*

In this section, we describe the evaluation methodology for the instruments and the gesture annotations.

### 2.5.1. Gesture Segmentation

To validate the gesture annotations presented in Section 2.3, we propose the recurrent neural network (RNN) model of Figure 13. Input data of the network are a sequence of kinematic data collected with the dVRK during the experiments. The ROSMA dataset includes 154 kinematics features from the dVRK platform (both master and slave sides). To isolate the gesture segmentation methodology from the particular robotic system employed to carry out the experiments, we have only considered PSM kinematics for gesture segmentation. To be able to replicate the model proposed in this work in a different scenario, we have also obviated the Cartesian position of the manipulators. Thus, for gesture segmentation we have used a subset of 34 kinematic features to train the network shown in Table 8. This subset includes tools orientation (given as quaternion), linear, and angular velocity, and wrench force and torque are raw data collected from the dVRK. Alongside these data, we have added two hand-crafted variables that provide useful information on the relation between the tools: distance and angle between PSM1 and PSM2. As gesture annotations on ROSMAG40, this dataset has been defined to be as generalizable as possible. Hence, no images have been added to the input of the network, as they would condition the learning process to a particular experimental scenario.

These 34 features are the input to the network of Figure 13. The sequence layer extracts the input from the given sequence. Then, a Bi-directional Long Short-Term Memory (Bi-

LSTM) layer with 50 hidden units learns long-term dependencies between time steps of the sequential data. This layer is followed by a dropout layer of 0.5 dropout rate to reduce overfitting. The network concludes with a 50-fully connected layer, a softmax layer, and a classification layer, which infers the output predicted gesture. LSTM layers are effective classifiers for time series sequence data, as they employ three control units, namely the input gate, the output gate, and the forget gate, to keep long-term and short-term dependencies. A bi-LSTM layer is a model with two LSTM networks that work in two directions: one LSMT layer takes the input in a forward direction, and the other in a backward direction. This model allows increasing the amount of information available to the network and improving the model performance [36].

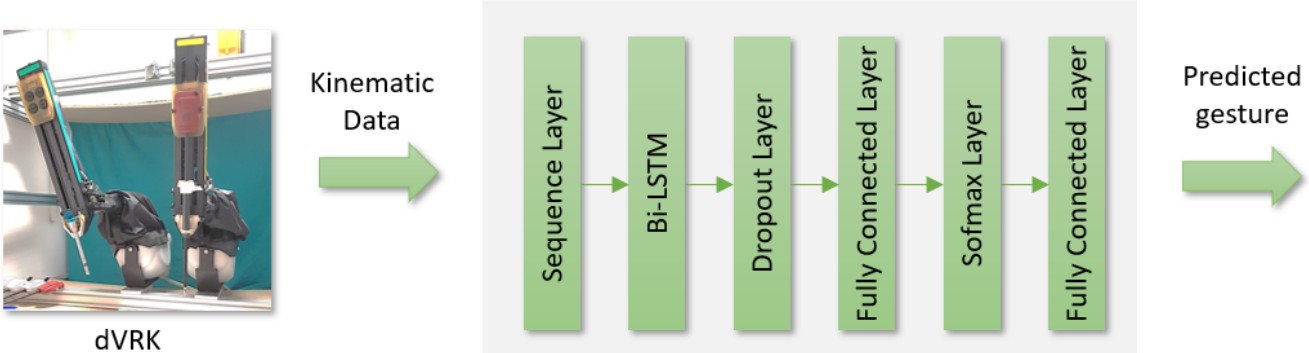

**Figure 13.** RRN model based on a bi-directional LSTM network for gesture segmentation.

**Table 8.** Kinematic data variables from the dVRK used as input to the RNN for gesture segmentation.

| Kinematic Variable | PSM | No. Features |
|---|---|---|
| Tool orientation (x, y, z, w) | PSM1 | 4 |
| | PSM2 | 4 |
| Linear velocity (x, y, z) | PSM1 | 3 |
| | PSM2 | 3 |
| Angular velocity (x, y, z) | PSM1 | 3 |
| | PSM2 | 3 |
| Wrench force (x, y, z) | PSM1 | 3 |
| | PSM2 | 3 |
| Wrench torque (x, y, z) | PSM1 | 3 |
| | PSM2 | 3 |
| Distance between tools | - | 1 |
| Angle between tools | - | 1 |
| Total number of input features | - | 34 |

The experimental setup includes four cross-validation schemes based on [8]:

- **Leave-one-user-out (LOUO)**: in the LOUO setup, we created five folds, each one consisting of data from one of the five users. This setup can be used to evaluate the robustness of the model when a subject is not seen by the model.
- **Leave-one-supertrial-out (LOSO)**: a supertrial is defined as in Gao et al. [8] as the set of trials from all subjects for a given surgical task. Thus, we created two folds, each comprising data from one of the two tasks. This setup can be used to evaluate the robustness of the method for a new task.
- **Leave-one-psm-out (LOPO)**: as half of the trials of the dataset are performed with PSM1 as the dominant tool while the other half are performed with PSM2 as the dominant tool, we have created two folds, one for trials of each dominant tool. This setup

can be used to evaluate the robustness of the model when tasks are not performed following a predefined order.

- **Leave-one-trial-out (LOTO)**: a trial is defined as the performance by one subject of one instance of a specific task. For this cross-validation scheme, we have considered the following test data partitions:

  – Test data 1: test data include two trials per user as follows: one of each task, and performed with a different PSM as dominant. Thus, we have left out 10 trials: 2 per user, 5 from each task, and 5 with one PSM as dominant. This setup allows us to train the model with the widest variety possible.

  – Test data 2: this test folder includes 10 trials of *pea on a peg* task, 2 trials per user with different PSM as dominant. This setup allows evaluating the robustness of the method when the network has significantly more observations of one task.

  – Test data 3: the same philosophy of test data 2, but leaving for testing just *post and sleeve* data for testing.

  – Test data 4: this test folder includes 10 trials performed with PSM1 as the dominant tool, and 2 trials per user and task. This setup allows evaluating the robustness of the method when the network has significantly more observations with a particular workflow of the task performance.

  – Test data 5: the same philosophy of test data 4, but leaving for testing just performance with PSM2 as the dominant tool.

### 2.5.2. Instrument Detection

In this work, YOLOv4 was used to detect the tip of the instruments, using the images as an input. YOLO (You Only Look Once) is a popular single-shot object detector known for its speed and accuracy. This model is an end-to-end neural network that makes predictions of bounding boxes and class probabilities all at once. This model is composed of three parts: backbone, neck, and head. The backbone is a pretrained CNN that computes feature maps from the input images. The neck connects the backbone and the head. It consists of a spatial pyramid pooling module and a path aggregation network, which merges the feature maps from various layers of the backbone network and forwards them as inputs to the head. The head processes the aggregated features and makes predictions for bounding boxes, objectness scores, and classification scores. Yolov4 has already proven to provide good results for surgical instrument detection [37].

The experimental setup for instrument detection includes two scenarios:

- **Leave-One-Supertrial-Out (LOSO)**: for this setup, we only used images of videos performing *pea on a peg* for training the network, and then we incorporated images of videos performing *post and leave* for testing. This setup can be used to evaluate the robustness of the method for different experimental scenarios.

- **Leave-One-Trial-Out (LOTO)**: for this setup, we used images of videos performing *pea on a peg* and *post and sleeve* for training and testing the network.

## 3. Results

This section presents the experimental results for the gesture segmentation network described in Section 2.5.1 and the instruments detection method presented in Section 2.5.2. All the experiments have been conducted on Intel(R) Xeon(R) Gold 5317 CPU @ 3.00 GHz with GA120GL (RTX A6000) GPU running Ubuntu 20.04.5 LTS. The code used to generate the experimental results, along with video demonstrations of the gesture segmentation network and the instrument detection model can be found in Appendix A and in the Supplementary Materials. These videos demonstrate the performance of the networks for the two algorithms presented in this work.

### 3.1. Results for Gesture Segmentation

This section presents the results of the gesture segmentation network described in Section 2.5.1. We present the results for the two categories of annotations of ROSGMAG40

dataset and the four experimental setups: leave-one-user-out (LOUO), leave-one-supertrial-out (LOSO), leave-one-psm-out (LOPO) and leave-one-trial-out (LOTO). Matlab 2023b software was used to implement the RNN model. Training of the network was performed with the Adam optimization algorithm with an initial learning rate of 0.001. We used a batch size of 8 for 60 epochs (240 iterations). A higher number of training epochs resulted in worse metrics due to overfitting of the network. The total number of observations of the network is 72,843. The size of the test data varies depending on the cross-validation method evaluated, ranging from 15,347 to 36,215. The detection time of the network ranges from 0.0021 to 0.0035 s.

Table 9 shows the results for LOUO cross-validation scheme. Observing these results we can deduce that the model suffers from a lack of generalization when a user is left out of the training. We can observe that the mean average precision varies from 39% for user *X01* to 64.9% for *X02*. Though the experimental protocol is the same for all users, these results suggest that the performance is highly dependent on the skill of each user. Moreover, we left freedom to complete the tasks in a random order, i.e., each user had to place six peas on top of the pegs and transfer the six colored sleeves from one side to the other of the pegboard, but the order in which they had to complete the task was not predefined.

**Table 9.** Results for Leave-One-User-Out (LOUO) cross-validation scheme.

| User Left Out Id | PSM1 mAP (MD) | PSM2 mAP (MD) | PSM1 mAP (FGD) | PSM2 mAP (FGD) |
|---|---|---|---|---|
| X1 | 48.9% | 39% | 46.26% | 23.36% |
| X2 | 58.8% | **64.9%** | 48.16% | 51.39% |
| X3 | 50.4% | 64.2% | 39.71% | 51.24% |
| X4 | **63.0%** | 61.2% | 54.05% | 49.08% |
| X5 | 54.6% | 53.6% | 52.34% | 52.7% |
| Mean | 55.14% | 56.58 | 48.01% | 45.55% |

Table 10 shows the results for the LOSO cross-validation scheme. As in the previous case, the model has difficulties extrapolating the features learned by the network to a task that has not been seen before. Table 11 shows the results for the LOPO cross-validation scheme. In this case, we can observe how the network has poor results for the low-level FGD annotations, but good results for high-level MD annotations, reaching a maximum mAP of 67.5%.

**Table 10.** Results for Leave-One-Supertrial-Out (LOSO) cross-validation scheme.

| Supertrial Left Out | PSM1 mAP (MD) | PSM2 mAP (MD) | PSM1 mAP (FGD) | PSM2 mAP (FGD) |
|---|---|---|---|---|
| Pea on a peg | 56.15% | 56% | 46.36% | 46.67% |
| Post and sleeve | 52.2% | 51.9% | 39.06% | 43.38% |

**Table 11.** Results for Leave-One-Psm-Out (LOPO) cross-validation scheme.

| Dominant PSM | PSM1 mAP (MD) | PSM2 mAP (MD) | PSM1 mAP (FGD) | PSM2 mAP (FGD) |
|---|---|---|---|---|
| PSM1 | 56.53% | 67% | 24.11% | 37.33% |
| PSM2 | **65.7%** | **67.5%** | 52.47% | 54.68% |

Table 12 shows the results for the LOTO cross-validation scheme. This is the experimental setup with more variety of the training data, thus it has the best results of the four cross-validation schemes. Using the test folds of test data 1, which offers the most variety of observations to the network, we reach a 64.65% and 71.39% mAP for the segmentation of PSM1 and PSM2 gestures, respectively, using MD annotations. However, for the segmentation of PSM1 gesture, we reach a maximum precision of 71.39% using test data 5 folds

and FGD annotations. To conduct a more comprehensive analysis of the reported results, we provide the confusion matrices, the training loss, and the training accuracy for this experimental setup using test folds of *Test data 1* in Appendix B.

**Table 12.** Results for Leave-One-Trial-Out (LOTO) cross-validation scheme.

| Dominant PSM | PSM1 mAP (MD) | PSM2 mAP (MD) | PSM1 mAP (FGD) | PSM2 mAP (FGD) |
|---|---|---|---|---|
| Test data 1 | 64.65% | **77.35%** | 56.46% | 58.99% |
| Test data 2 | 63.8% | 62.8% | 30.43% | 70.58% |
| Test data 3 | 53.26% | 55.58% | 53.26% | 61.6% |
| Test data 4 | 48.72% | 60.62% | 58.3% | 55.58% |
| Test data 5 | 60.51% | 66.84% | **71.39%** | 46.67% |

### 3.2. Results for Instruments Detection

We present experimental results for two architectures of the YOLOv4 network: a CSPDarkNet53 backbone pretrained on the COCO dataset, and the compressed version YOLOv4-tiny. We trained both networks for 40 epochs with the Adam optimizer with a momentum of 0.9 [38], a batch size of 16, and an initial learning rate of 0.001. The learning rate is divided by a factor of 10 every 10 epochs.

Table 13 shows the experimental results for the LOSO experimental setup. Both models have high-precision results for the detection of both tools, reaching over 80% mAP. This reveals the generalization capabilities of the YOLOv4 network to detect the instruments in a scenario the network has not seen before. On the other hand, Table 14 presents the experimental results for the LOTO experimental setup. As expected, results when the network has seen both task scenarios during the training are higher, reaching values of 97.06% and 95.22% mAP for PSM1 and PSM2, respectively. Comparison between the performance using CSPDarket53 and YOLOv4-tiny shows that the tiny version provides similar accuracy results but the performance is 2/3 times faster. Figure 14 shows the precision-recall curves for this model, showing both high recall and high precision for the PSMs detection.

**Table 13.** Results for LOSO experimental setup.

| Architecture | Test Data | PSM1 mAP | PSM2 mAP |
|---|---|---|---|
| CSPDarknet53 | Post and sleeve | 70.92% | 84.66% |
| YOLOv4-tiny | Post and sleeve | 83.64% | 73.45% |

**Table 14.** Results for LOTO experimental setup.

| Architecture | Left Tool mAP | Right Tool mAP | Detection Time |
|---|---|---|---|
| CSPDarknet53 | **97.06%** | **95.22%** | 0.0335 s (30 fps) |
| YOLOv4-tiny | 93.63% | 95.8% | 0.02 s (50 fps) |

Figure 15 shows different examples of correct (top images) and incorrect (bottom images) tool detection. In this figure, we can observe two examples of how the model is able to detect and distinguish the instruments even when they cross and there is an overlap of their bounding boxes. This figure also shows the more representative situations of incorrect detections, i.e., when the network detects more than two instruments in the image, and incorrect detection when the instruments cross.

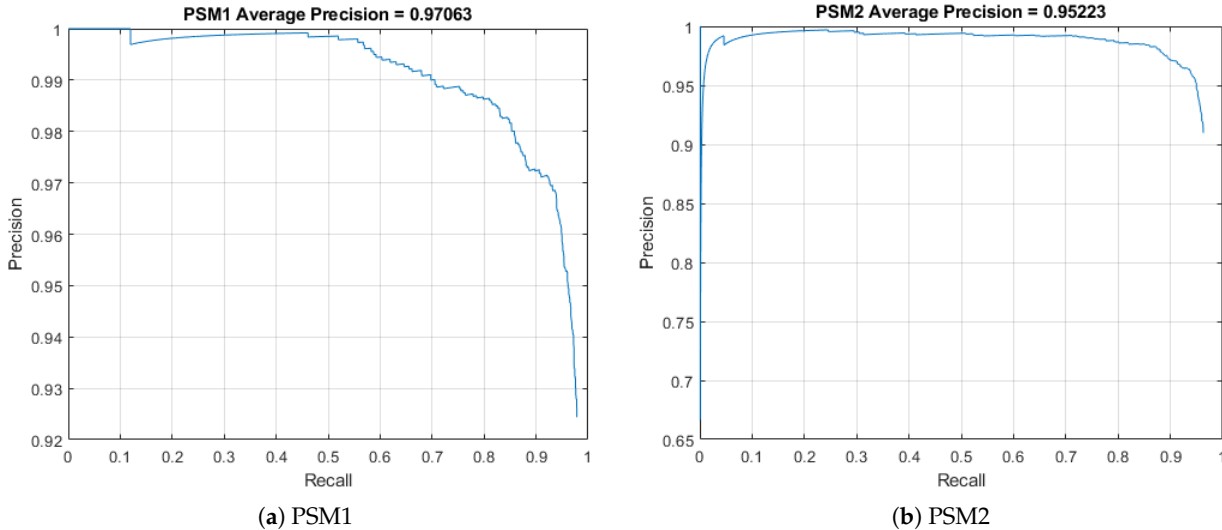

(**a**) PSM1      (**b**) PSM2

**Figure 14.** Precision-recall curve for the LOTO cross-validation scheme using CSPDarknet53 architecture.

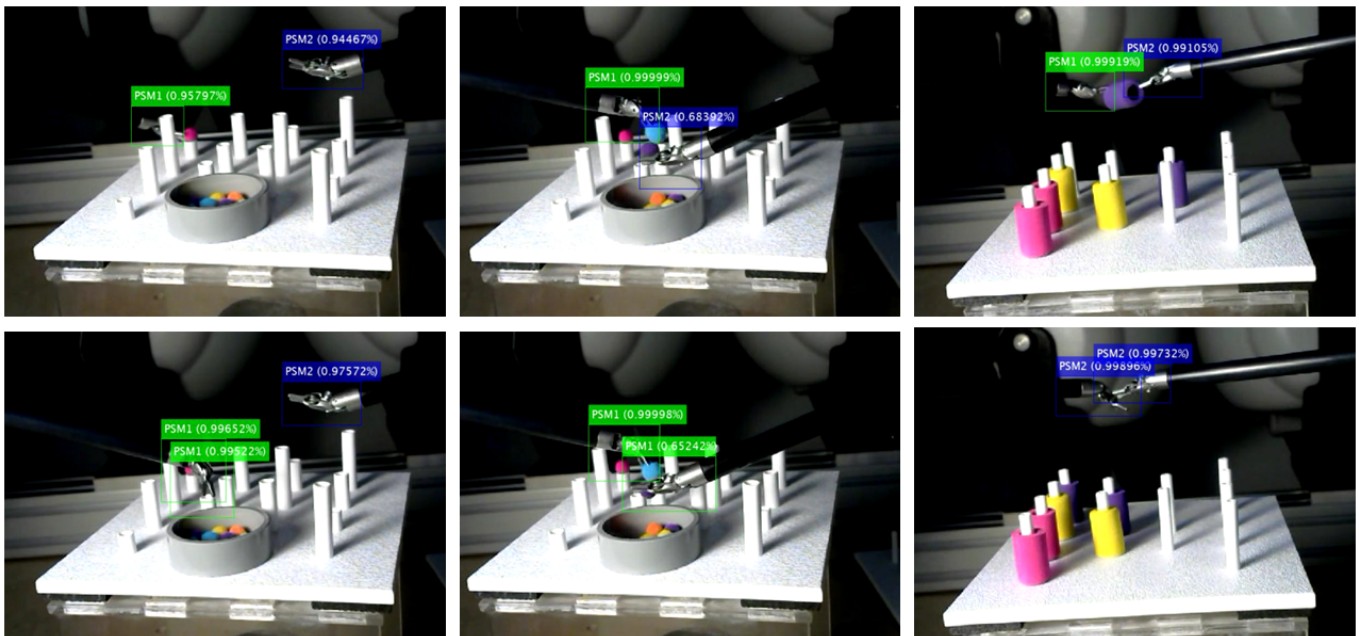

**Figure 15.** Example of correct detections (**top** images) and incorrect detection (**bottom** images).

## 4. Discussion

This work explores the approach of studying the motion of the surgical instruments separately one from each other. In this sense, instead of considering a gesture or maneuver as a part of an action that involves the coordinated motion of the two tools the surgeon is managing, we define gestures as actions each tool is performing independently, whether they are interacting with the other tool or not. We consider that this approach would facilitate the generalization of the recognition methods for procedures that do not follow rigid protocols. To train the recurrent proposed neural network we have used kinematic data without Cartesian position to allow the reproduction of the experiments with different robotic platforms. We have decided not to use images as an input to the network to be able to extrapolate the results to different scenarios, i.e., the idea is that the network learns behavioral patterns of the motion of the tools, whether they are picking colored sleeves, peas, rings, or any other object.

Results reveal a high dependency of the model on the user skills, with a wide range of precision from 39% to 64.9% mAP depending on the user left out for the LOUO cross-validation scheme. In future works, we will investigate the model performance when it is trained with a higher variety of users. The model also has a high dependency on the task used for training but shows high robustness for changes in the tool. The mean accuracy of the model is over the 65% when the model has been trained using a particular tool as the dominant one to complete the task, but tested with trials with a different dominant tool. These are promising results to generalize the recognition method for dexterous or left-handed surgeons.

When the system is trained with a wide variety of trials, comprising different users, tasks, and dominant tools, the performance of the gestures prediction reaches 77.3% mAP. This result is comparable to other works that perform gesture segmentation using kinematics data. Luongo et al. [39] achieved 71% mAP using only kinematic data as an input to a RNN on the JIGSAWS dataset. Other works that include images as input data report results of 70.6% mAP [25] and 79.1% mAP [26]. Thus, we consider that we achieved a good result, especially taking into account that recognition is performed on tasks that do not follow a specific order in any of the attempts. We believe that the annotations provided on ROSMAG40 are a good base to advance in the generalization of the gesture and phase recognition methodologies for procedures with a non-rigid protocol. Moreover, the annotations presented in this work could be merged with the traditional way of annotating surgical phases to provide low-level information on the performance of each tool, which could improve the high-level phase recognition with additional information.

The annotations provided in ROSMAT24 for tool detection can be used as complementary to the gesture segmentation methods to focus the attention on a particular tool. This is important because surgeons usually perform the accuracy tasks with their dexterous hand and the support tasks with the non-dexterous one. Thus, being able to detect each one independently can provide useful information. We have demonstrated that the YOLOv4 network provides high precision in the instrument detection, reaching 97% mAP. This result is comparable to other works on surgical instruments detection, such as Kurman et al. [7], who reported a 90% mAP or Zhao et al. [40] with 91.6% mAP. We also demonstrated the capabilities of the network to detect the instruments in an unseen scenario through the LOSO experimental setup.

**Supplementary Materials:** The following supporting information can be downloaded at https://www.mdpi.com/article/10.3390/app14093701/s1, Videos: Demonstrations of the networks performance.

**Author Contributions:** Methodology, I.R.-B.; writing—original draft preparation, I.R.-B.; writing—review and editing, C.L.-C.; validation, J.M.H.-L. and I.R.-B.; formal analysis, J.M.H.-L.; software, J.C.-V., J.M.H.-L., I.R.-B. and C.L.-C.; data curation, J.C.-V.; supervision, C.J.P.-d.-P. All authors have read and agreed to the published version of the manuscript.

**Funding:** This research was funded by the Spanish Ministry of Science and Innovation under grant number PID2021-125050OA-I00.

**Institutional Review Board Statement:** Not applicable.

**Informed Consent Statement:** Not applicable.

**Data Availability Statement:** The code supporting the reported results can be found at (https://github.com/irivas-uma/rosma, accessed on 26 February 2024), and the data are publicly available at the Zenodo website for both ROSMAT24 (https://zenodo.org/records/10721398, accessed on 26 February 2024) and ROSMAG40 (https://zenodo.org/records/10719748, accessed on 26 February 2024) datasets.

**Conflicts of Interest:** The authors declare no conflicts of interest.

**Abbreviations**

The following abbreviations are used in this manuscript:

| | |
|---|---|
| ROSMA | Robotics Surgical Maneuvers |
| dVRK | da Vinci Research Kit |
| MD | Maneuver Descriptor |
| FGD | Fine-Grained Descriptor |
| RNN | Recurrent Neural Network |
| CNN | Convolutional Neural Network |
| Bi-LSTM | Bidirectional Long-short Term Memory |
| MT | Master Tool Manipulator |
| PSM | Patient-sided Manipulator |
| YOLO | You Only Look Once |

**Appendix A**

The code used to generate the results presented in this work is provided on GitHub: https://github.com/irivas-uma/rosma, accessed on 26 February 2024. Besides the code, this repository also contains video demonstrations of the instrument detection and the gesture segmentation algorithms presented in this work. The content of these videos is as follows:

- Testgestures_peaonapeg.mp4: this video shows the annotated labels for a trial of the *pea on a peg* task using the FGD annotations. Gesture of PSM1 is displayed in green on the left side of the image, and gesture of PSM2 is displayed in blue on the right side of the image. The purpose of this video is to provide the reader with a better understanding of the meaning of the gestures.
- Testgestures_postandsleeve.mp4: this video shows the performance of the gesture recognition network for a trial of the *post and sleeve* task using the MD annotations. The predicted gesture of PSM1 is displayed in green on the left side of the image, and the predicted gesture of PSM2 is displayed in blue on the right side of the image. The purpose of this video is to demonstrate the performance of the network.
- ToolsDetection_peaonapeg.mp4: this video shows the performance of the instruments detection network for a trial of the *pea on a peg* task. The predicted bounding boxes for PSM1 and PSM2 are shown in green and blue, respectively, with a label on the top displaying the predicted accuracy.
- ToolsDetection_postandsleeve.mp4: performance of the instruments detection network for a trial of the *post and sleeve* task. The predicted bounding boxes for PSM1 and PSM2 are shown in green and blue, respectively, with a label on the top displaying the predicted accuracy.

**Appendix B**

Figures A1 and A2 show the confusion matrices for MD and FGD gesture segmentation for the particular experiment of the LOTO cross-validation method using the test data partition labeled as *Test data 1*. On these confusion matrices, the rows correspond to the predicted class (*Output class*) and the columns correspond to the true class (*Target class*). The diagonal cells correspond to observations that are correctly classified. The off-diagonal cells correspond to incorrectly classified observations. The number of observations and the percentage of the total observations are shown in each cell. The column on the far right of the plot shows the percentages of all the examples predicted to belong to each class that are correctly and incorrectly classified. The row at the bottom of the plot shows the percentages of all the examples belonging to each class that are correctly and incorrectly classified. The cell in the bottom right of the plot shows the overall accuracy.

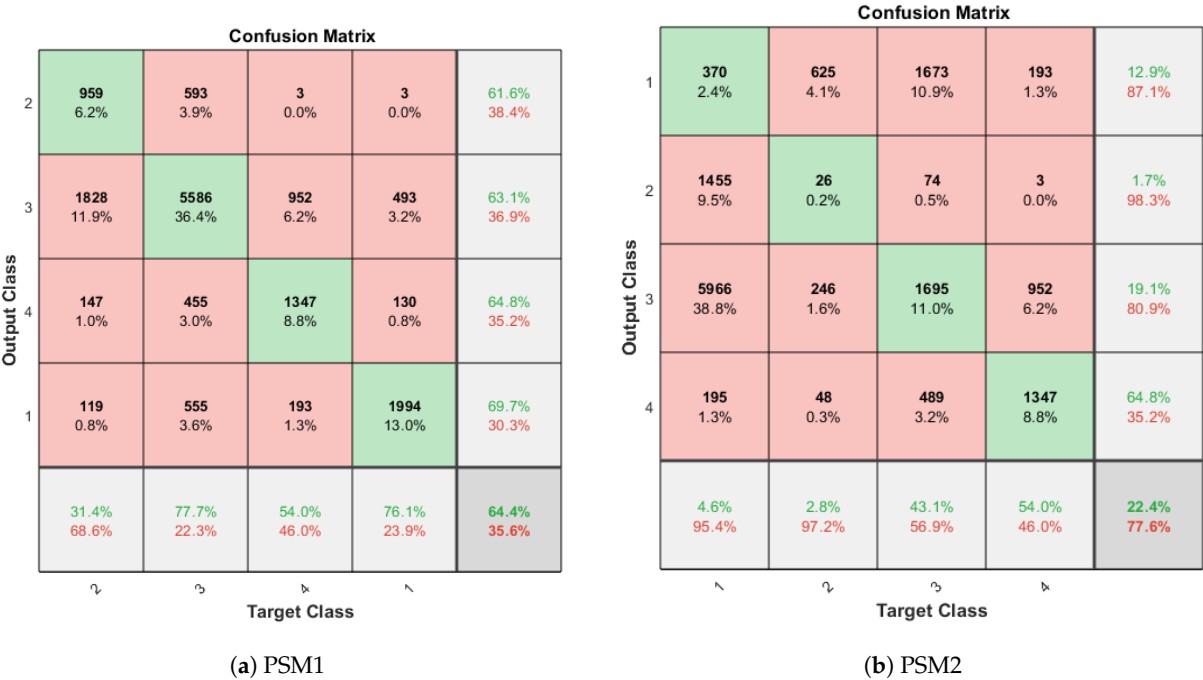

(**a**) PSM1 (**b**) PSM2

**Figure A1.** Confusion matrix for gesture segmentation using MD annotation, for the LOTO cross-validation scheme using the test data partition labeled as *Test data 1*.

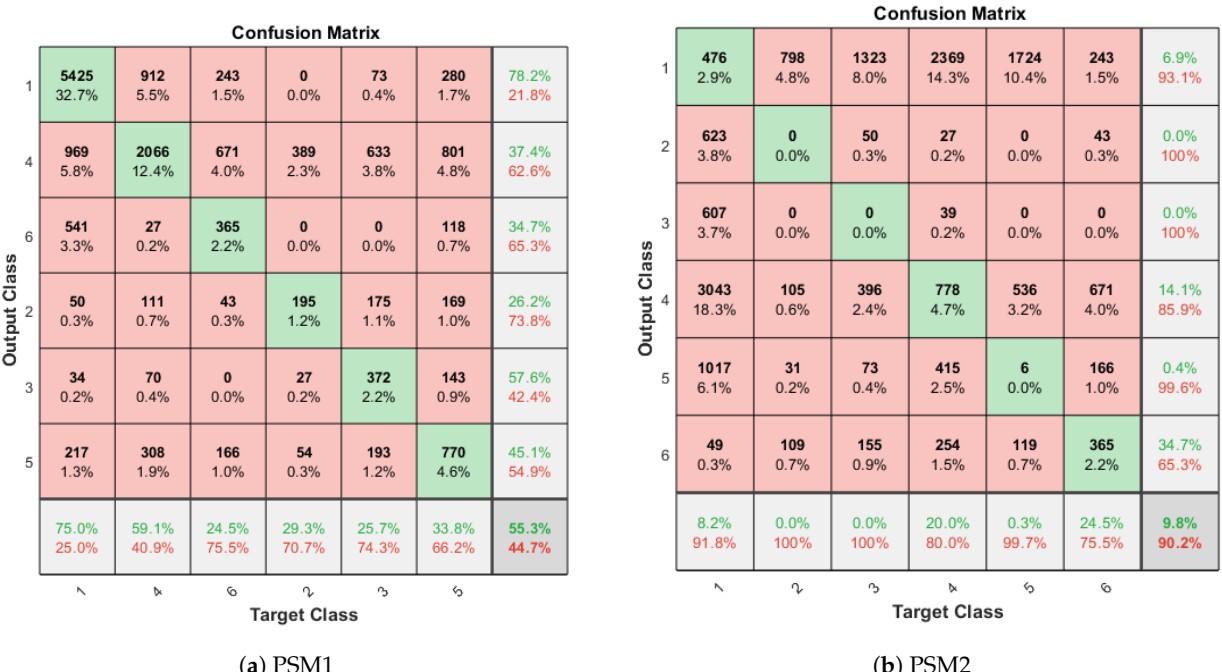

(**a**) PSM1 (**b**) PSM2

**Figure A2.** Confusion matrix for gesture segmentation using FGD annotation for the LOTO cross-validation scheme using the test data partition labeled as *Test data 1*.

On the other hand, Figures A3 and A4 represent the training accuracy and the training loss for the LOTO cross-validation scheme using the test data partition labeled as *Test data 1* for MD and FGD annotations, respectively. The results for PSM1 gesture training are shown in blue, while the results for PSM2 gesture training are shown in red.

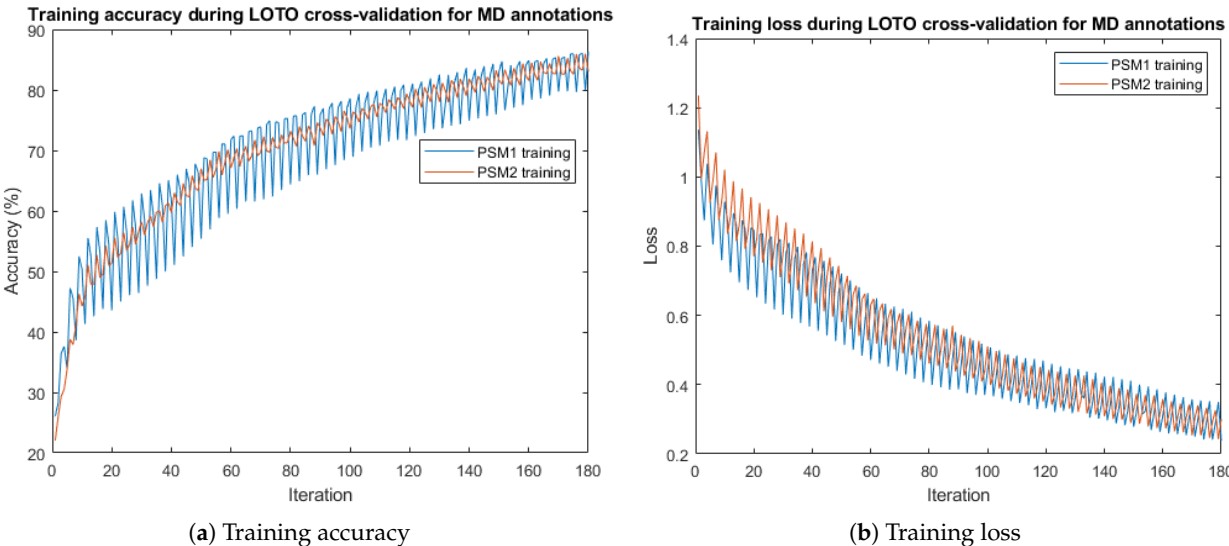

(**a**) Training accuracy         (**b**) Training loss

**Figure A3.** Representation of the training accuracy (**a**) and the training loss (**b**) for the LOTO cross-validation scheme using the test data partition labeled as *Test data 1* and MD annotations.

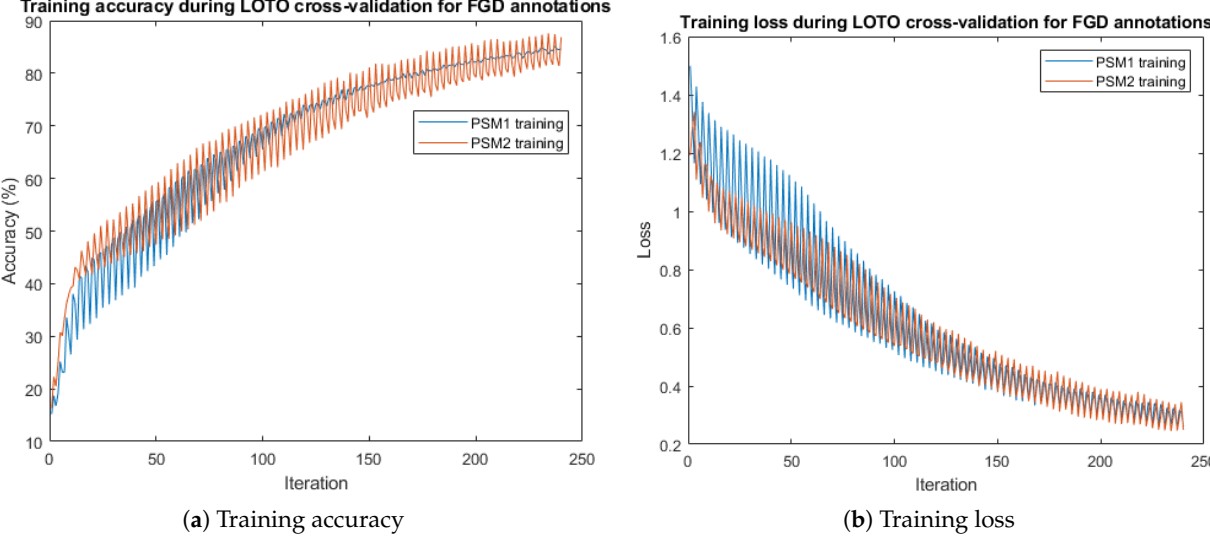

(**a**) Training accuracy         (**b**) Training loss

**Figure A4.** Representation of the training accuracy (**a**) and the training loss (**b**) for the LOTO cross-validation scheme using the test data partition labeled as *Test data 1* and FGD annotations.

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
