# Peer review of "Instrument Detection and Descriptive Gesture Segmentation on a Robotic Surgical Maneuvers Dataset"

_applsci, doi:10.3390/app14093701_

Round 1
Reviewer 1 Report
Comments and Suggestions for Authors
The authors have introduced a new datasets which are a subset of RASMA dataset. The datasets try to fill a significant information which can be very impactful in subsequent research. I congratulate the authors for recognizing the gap in information and trying to address this. The authors then tried to use the recorded dataset for instrument detection and gesture recognition, both of which are necessary steps for robotic surgery. The authors have tried to do these in a subject independent manner so that the algorithm is unaffected by the person operating the robot, which in my opinion is a commendable effort.
The first half of the paper can use some extra and detailed explanations and necessary introduction to some critical background information to improve the overall quality, especially the platform details used for data collection requires crucial explanation without which it becomes very difficult to understand the data collection method. This is a critical shortcoming of the paper.
Some other ideas which need to be corrected or elaborated are as follows:
While describing the ROSMA dataset, it is mentioned that dataset has 36 kinematic variables and 154 dimensional data. The same has been used in the new datasets. The table 6 mentions only 34 variables. There is no mention on the details of 156 dimensions.
The ROSMA dataset has 3 tasks but the new dataset records data from 3 tasks and does not detail about the why the third task was not included in this work.
Through out the text, the authors have used decimal point instead of comma as the separator for writing large numbers. For example 72,843 is written as 78.843 .
The authors have not provided a truth table for any of the classification tasks. This is a must to understand the results properly. This can be added as supplementary infomation.
From figure 8, it appears that the data is biased as it has a lot more data points for "idle" as compared to others. How was this bias handled?
The approach using two types of descriptors i.e. MD and FGD is a little confusing. 2 of the 4 gestures in MD are also a part of FGD. Would it not be more logical to make FGD as a subcase of MD. The logic of using distinct descriptors must be justified.
The method of annotating the images for instrument detection is not described in details. Was it manual or semi-automatic. Was it frame by frame? How were the bounding box size determined.
What were the final errors for training and validation datasets. An analysis of these can shed light on if more data will be helpful in improving the performance.
It would be crucial to know what percentage of robots workspace was used to collect the data. Especially when the robot kinematics are the input data, this consideration should not be neglected.
Author Response
Dear reviewer,
we appreciate your comments on our work. You can find a response to all your suggestions in the document attached.
Best regards.

Reviewer 2 Report
Comments and Suggestions for Authors
Dear Authors,
I read your work, "Instrument Detection and Descriptive Gesture Segmentation on a Robotic Surgical Maneuvers Dataset," with great interest.
In the era of Artificial Intelligence, machine learning, and robotic surgery, your study addresses a highly relevant topic. The authors have developed a neural network based on a bidirectional long-short term memory layer, validated through four experimental setups.
The work is well-designed, and the methodology appears robust. The statistical analysis is appropriate. I particularly appreciate the completeness of the introduction and the proper development of all sections.
I believe this work, while not exceptionally novel, holds sufficient merit for consideration for publication.
I could not identify any significant issues; however, I suggest a minor English revision.
I am at your disposal.
Best Regards
Comments on the Quality of English LanguageMinor revision suggested
Author Response
Dear reviewer,
we appreciate your comments on our work. Following your suggestions, we have revised the English throughout the text.
Best regards.
Reviewer 3 Report
Comments and Suggestions for Authors
The submission propose to extend a known dataset by means of two datasets, one where bounding box annotations. for instrument detection are included and another one for high and low level gesture annotations.
I consider that many concepts are taken for granted, i.e, as already known and this is not necessarily the case in the readers. Thus...
1. Please start defining the term "annotations" in the sense of this field (you start to use the term from the abstract, but you never say what it is.
2. In this sense, with all respect, explain properly how/why you apply these annotations in the submission. For instance, if you try to detect a chirurgical instrument, given a set of kinematic trajectories, why not to use a control systems algorithm or an identification systems method? Explain.
3. According to (2) and from [33] (where numerical databases are non-readable unless you import it) how are the trajectories defined? How many degrees of freedom do you consider in the manipulators; i.e., in the grippers?
I mean, does (say) x1(t) is the trajectory of the wrist, x2(t), finger1 ,etc ? Explain how/who are these trajectories defined.
4. How were those trajectories obtained? It is obviously mentioned that from <<a kinematic model>> (missing by the way) but how to interpret the databases of [33]?
5. Related to (4): Besides analyzing the numerical data, with all respect, from the provided videos there, I can not see the "annotations".
6. Please include all these explanations in the Abstract and in the Intro.
7. I find the videos given in [33] interesting and you should improve the document to remark these facts.
8. You present boxplot diagrams but given that you execute many tests you should include a statistical part where confidence intervals could be added to offer more support to this work.
9. Why not to work with dynamical datasets (the opposite to kinematic data)?
10. In line 421 and following, authors say that :"We consider that this approach would facilitate the generalization of the recognition methods for procedures that do not follow rigid protocols. To train the recurrent neural network proposed we have used kinematic data without cartesian position to allow the reproduction of the experiments with different robotic platforms.". How is it possible that without a cartesian reference the reproduction of the experiments are possible? This is opposite to the paragraph that starts at line 282 and following (including table 6) where you actually give mechanical variables in terms of (x,y,z) [and by the way, in the first row, you give (x,y,z,w)?? do you live in a 4th dimensional space..? I don't...
11. Related to (10) and (3), that is why I ask what is the meaning of the trajectories presented in [33] and mentioned/used here. In addition, you claim that cartesian coordinates are not necessary (??) but you do not work in any other reference frame (spherical, cylindrical ...etc). Use of mechanical variables without reference frames????
12. Please also improve the mechanical part and the relation with numerical data.
Author Response

(The authors gave the same response as above.)

Round 2
Reviewer 1 Report
Comments and Suggestions for Authors
The authors have successfully answered all my comments and I have no further questions.
Author Response
Thank you very much.
Reviewer 3 Report
Comments and Suggestions for Authors
Please not only highlight the text you modified but also include some type of guide where you clearly indicate that the comments were addressed.
Comments on the Quality of English Language
None.
Author Response
Dear reviewer,
we uploaded a pdf file with our responses to all your comments. We attach the document again, and we also copy the responses here in case there is a problem with the upload.
Thank you for your comments. The paper has been improved following your comments and suggestions. You can find our responses to your comments below
- Please start defining the term "annotations" in the sense of this field (you start to use the term from the abstract, but you never say what it is.
We have included a general definition of annotation at the beginning of the introduction (line 45).
- In this sense, with all respect, explain properly how/why you apply these annotations in the submission. For instance, if you try to detect a chirurgical instrument, given a set of kinematic trajectories, why not to use a control systems algorithm or an identification systems method? Explain.
We use these annotations as input of neural networks. In the case of instruments detection, we do not use the kinematic data, we only use images as input to the network. We have clarified this in section 2.5.2 (line 382). For gesture segmentation we use the kinematic data as input to the RNN model. We have decided to use a neural network as segmentation method because it has been proven in the literature that this type of algorithms provide good results.
- According to (2) and from [33] (where numerical databases are non-readable unless you import it) how are the trajectories defined? How many degrees of freedom do you consider in the manipulators; i.e., in the grippers?
I mean, does (say) x1(t) is the trajectory of the wrist, x2(t), finger1 ,etc ? Explain how/who are these trajectories defined.
A detailed description of the dVRK platform and the data and the recording methodology are reported in our previous work [29]. However, to facilitate the understanding of the work presented in this paper, we have added a section (Section 2.1) providing a detailed description of the dVRK platform, including the degrees of freedom of the manipulators, the kinematics, and the reference frames. Moreover, we have added information about the tasks and the data in Section 2.2. Table 1 includes a detailed description of the kinematic variables recorded in the dataset.
- How were those trajectories obtained? It is obviously mentioned that from <<a kinematic model>> (missing by the way) but how to interpret the databases of [33]?
A detailed description of the data is provided has been included in Table 1. The description of the kinematic model of the platform has been included in Section 2.1, in particular, Figure 2 shows the reference frames of the components of the system.
- Related to (4): Besides analyzing the numerical data, with all respect, from the provided videos there, I can not see the "annotations".
To clarify the content of the datasets, we have described the directory structure for ROSMAG40 (in section 2.3.1, line 214) and for ROSMAT24 (in section 2.4, line 310). The videos of these dataset are the raw videos recording during the experiments. The annotation for gesture segmentation are in the folders MDlabels and FGDlabels. Each row of these files corresponds with a video frame. In the case of instruments annotations, the labels folder contains the bounding boxes for each frame of the videos. The annotations overlapped on the videos are shown in the videos given in Appendix A, where we show the performance of our models after training the networks. A detailed description of the content of these videos is given in line 532. Readers may see the performance of the network on other video different from the ones provided by the authors using the matlab code provided in the link to our GitHub repository (line 530).
- I find the videos given in [33] interesting and you should improve the document to remark these facts.
We appreciate your comment. We have added a sentence highlighting the importance of these videos in section 3 (line 406), and we have also included a description of the content of the videos in Appendix A (line 532).
- You present boxplot diagrams but given that you execute many tests you should include a statistical part where confidence intervals could be added to offer more support to this work.
Figures 5 and 9 have been changed following this suggestion.
- Why not to work with dynamical datasets (the opposite to kinematic data)?
In our datasets, we have included wrench force, wrench torque and joint efforts of the patient side manipulators, as it can be seen in Table 8. This is the only force data available from the experimental platform.
- In line 421 and following, authors say that :"We consider that this approach would facilitate the generalization of the recognition methods for procedures that do not follow rigid protocols. To train the recurrent neural network proposed we have used kinematic data without cartesian position to allow the reproduction of the experiments with different robotic platforms.". How is it possible that without a cartesian reference the reproduction of the experiments are possible? This is opposite to the paragraph that starts at line 282 and following (including table 6) where you actually give mechanical variables in terms of (x,y,z) [and by the way, in the first row, you give (x,y,z,w)?? do you live in a 4th dimensional space..? I don't...
With the sentence “to allow the reproduction of the experiments with different robotic platforms”, we mean to be able to replicate the experiment with other robotic system using the same experimental board, which is a commercial one. Then, it would be possible to use these recording data with our network to predict the instruments gestures, as data is not affected by the position of the board with respect to the robots.
The orientation has four parameters because it is given as a quaternion. It has been clarified in line 331.
- Related to (10) and (3), that is why I ask what is the meaning of the trajectories presented in [33] and mentioned/used here. In addition, you claim that cartesian coordinates are not necessary (??) but you do not work in any other reference frame (spherical, cylindrical ...etc). Use of mechanical variables without reference frames????
Although it was described in our previous work [29], we have included Section 2.1 with the description of the system. In this section we have included a figure (Fig. 2) with the manipulators’ kinematics. Here, readers can see the reference frame of the system. Data of the two patient side manipulators is refer with respect to the common frame {ECM}.
- Please also improve the mechanical part and the relation with numerical data.
It has been improved in Section 2.1.
